# Experiences of the Initial Phase Implementation of the STAMINA-Model in Perioperative Context Addressing Environmental Issues Systematically—A Qualitative Study

**DOI:** 10.3390/ijerph17093037

**Published:** 2020-04-27

**Authors:** Erebouni Arakelian, Therese Hellman, Magnus Svartengren

**Affiliations:** 1Department of Surgical Sciences, Uppsala University, and AnOpIVA, Uppsala University Hospital, 751 85 Uppsala, Sweden; 2Department of Medical Sciences, Occupational and Environmental Medicine, Uppsala University, 752 36 Uppsala, Sweden; therese.hellman@medsci.uu.se (T.H.); magnus.svartengren@medsci.uu.se (M.S.)

**Keywords:** nursing, management, perioperative, qualitative, implementation, STAMINA model

## Abstract

(1) Background: Challenges in perioperative settings put demands on staff working with systematic work environment management. A support model, STAMINA (Structured and Time-effective Approach through Methods for an Inclusive and Active working life), was implemented in a hospital in Sweden, to help staff address environmental issues systematically. The aim was to describe the experiences of the initial phase of implementation of the adapted STAMINA model in perioperative context. (2) Methods: Qualitative individual interviews were held with 14 managers and employees (three men and 11 women). Data were analysed by systematic text condensation. (3) Results: Five themes were identified: Limited knowledge of the model and the implementation process; scepticism, lack of confidence in the model and a passive attitude; the model offered increased participation; the culture in the organization—to understand one’s role as employees and managers; and endurance and feedback are key factors for success in the implementation process. (4) Conclusions: Scepticism turned to positive attitude by recognising that the STAMINA model offered increased participation. In order to have successful implementation, the organisational culture must be taken into consideration by giving the employees increased responsibilities and timely feedbacks. Role description, goal definition, feedback, and sticking to one model are key factors for success.

## 1. Introduction

In Sweden, according to regulations, both employees and their managers have an obligation to create and maintain a healthy work environment. In health care, particularly in perioperative settings (operating departments, and pre- and post-operating departments), the work environment is special as it includes a high-tech environment, and a poor work environment affects employees [1] and the quality of patient care [2]. Despite provisions, there are still organisations that do not properly perform their systematic work environment management [3]. A specific support model, the STAMINA (Structured and Time-effective Approach through Methods for an Inclusive and Active working life) model, which aims to concretise the systematic work environment management [4], was implemented in the perioperative setting in a university hospital in central Sweden. This study investigates the experiences of the initial implementation process.

### 1.1. The Perioperative Environment and Its Demands

The environment in perioperative settings may be viewed as a complex organisation. The environment offers specialist care in a high-tech environment. Thus, the demands on the employees’ physical and psychosocial environment are high. Work conditions are unique as the specialist nurses in the perioperative settings are confined to the operating rooms with the possibility of having small breaks, which allows for minimum recovery time between two patients [1]. Having time for meetings and competence development is difficult due to the ongoing organisation around-the-clock. The national and global shortage of specialist nurses [5,6] makes patients vulnerable, as planned surgeries may be cancelled. These factors affect work environment, quality of care [2], and work satisfaction [1,7] among employees. Vowels et al. [8] indicated that operating room nurses experienced “pressure to work quickly” as a stressful event in their everyday work where patient turnover is high. According to Logde, Rudolfsson, Broberg, Rask-Andersen, Walinder, and Arakelian [1], specialist nurses in perioperative settings (i.e., nurse anaesthetists and operating room nurses) leave their work due to rough psychosocial environment and systematic freezing out colleagues, poor wage growth, dissatisfactory work schedule, working night and day shifts, and a nurse manager who is described as absent, non-including and “nonchalant”. Furthermore, Lee et al. [9] and Sillero and Zabalegui [10] indicated that emotional exhaustion, depersonalization, and reduced personal accomplishment were found among nurses the in perioperative context. The study by Lee, MacPhee, and Dahinten [9] pointed to nurse manager ability, leadership, and support of nurses, staffing, and resources adequacy; and nursing foundations of quality care as factors affecting the occurrence of burnout. Moreover, in the perioperative context, physician to nurse violence [11] and nurse to nurse violence [12] among operating room nurses were described earlier. Thus, nurse managers have their own challenges due to the specific characteristics of the perioperative settings as described in this paragraph, and due to the extensive responsibility they carry towards their employees and the patients [13]. Thus, there are several challenges in finding ways of working with the work environment systematically in the perioperative context.

### 1.2. Regulations and the Responsibility of Managers and Their Employees

In Sweden, employers are responsible for ensuring a good, healthy, work environment for their employees and excellent care for their patients [14]. Employers have the responsibility of preventing accidents and ill health at work, and to promote a satisfactory work environment, which is regulated by the provisions of systematic work environment management (AFS 2001:1) [15]. It is also the employers’ responsibility to investigate, carry out, and follow-up on activities connected to the work environment (AFS 2001:1) [15]. In 2016, a certain provision focusing on the organizational and social work environment was launched by the Swedish Work Environment Authority (SWEA) [16]. Managers and employees in perioperative settings are facing several challenges related to work environmental issues, many focusing on organisational issues such as sufficient time for recovery. Even though, it is known that work environmental management is insufficient provided in Sweden [3].

### 1.3. Challenges of Implementation

There are various reasons why systematic work environment management is not being properly performed; for example, it is known that interventions that do not produce rapid results tend to fail in the long-term implementation [17]. Furthermore, it is a challenge to find a suitable intervention for a specific workplace as intervention works differently in various contexts [18,19]. Thus, it is not surprising that interventions found to be effective in research fail to translate into meaningful outcomes across contexts in practice. Hojberg et al. [20] have identified several essential implementation components to optimise the implementation of workplace improvement initiatives. Eleven components were divided into four domains: a supportive organisational platform, an engaged workplace with mutual goals, an intervention adapted to the workplace, and an intervention that is seen as an attractive choice. However, it is still evident that there are knowledge gaps between evidence and practice regarding understanding of implementation components to facilitate work environment initiatives at the workplaces [21]. To practically facilitate the work with environmental issues, the STAMINA model was implemented in the perioperative context in a hospital in Sweden.

### 1.4. The STAMINA Model

The STAMINA (Structured and Time-effective Approach through Methods for an Inclusive and Active working life) model promotes employee participation and increased understanding of the operations in the organisation [22]. It is influenced by the structure of an organisational intervention [23] and the integrated model of group development [24,25]. The original STAMINA model consists of sessions delivered three times annually, in which the work groups focus on: (1) shared basic values, aims, and goals of the work group; (2) the work group’s current work situation; (3) how the work group wants their work situation to be; and (4) what actions can be taken to create the desired work situation. In the last step, the work group prioritises one activity they want to focus on and creates an action plan. The operational work based on the action plans is ongoing between the sessions. To make the model independent from external support, internal support is provided by facilitators within the organisation, who were educated about the model and implementation process. Even though the model holds a specific structure, it allows for adaptations to fit the actual workplace where it is to be implemented. Key features of the model are structure, recurrent feedback, and employee participation [4]. There are various challenges in implementing the model perioperative context in a hospital in Sweden.

In summary, there are provisions as well as work environment interventions in place to address the challenges in the work environment. Still, not all organisations pay appropriate attention to these questions. The aim of this study was to investigate experiences of the initial phase of implementation of the STAMINA model in the perioperative context.

## 2. Materials and Methods

### 2.1. Study Design

This study is a continuation of a larger project conducted in municipalities in Sweden. A description of the larger project is to be found in a prospective designed protocol [4]. This protocol also includes a detailed description of the STAMINA model. In the current study, after ethical approval by the Swedish Ethical Review Authority, a qualitative prospective study design was conducted in the new context of perioperative settings to capture the experiences of the self-sustaining implementation process of the support model and how it was perceived that the model contributed to the goal of a better working environment.

#### Study Setting

The study was performed in the perioperative section, with approximately 900 employees, at a university hospital in Sweden. The attrition rate of the nurses in the hospital under study was lower compared to other university hospitals in Sweden, but that the percentage of personnel who resigned was higher in comparison to others.

The STAMINA model was implemented and used by about 500 employees working in seven Operating Departments (OD), one postoperative department, one sterile processing department, three groups of anaesthesiologists, two groups of first line managers, and a head of department with her immediate subordinate leaders.

The working conditions in the perioperative settings are unique. The operating rooms are closed rooms, sometimes without access to daylight. Patient safety and hygiene reasons lead to specialist nurses, OR-nurses, not leaving the operating room while working with the patient. Patient care demands high concentration. During day shifts, only small lunch break of 30–45 min and two small coffee breaks of 10–15 min each are taken by the specialist nurses. OR-nurses usually take their breaks in between two patients. High demands are placed on both specialist nurses because of these working conditions during long hours of work [1].

### 2.2. The Implementation

After a reorganisation in the perioperative setting, the STAMINA model was chosen for implementation. Several educational events were arranged for the nurse managers and employees within the organisation. Two supervisors were chosen from the organisation to offer support during the implementation process. The model was modified and adjusted to the conditions of the perioperative setting to have an initial and longer (i.e., two hour) workshop, and then a shorter workshop (i.e., one hour), once during autumn (August until mid-January) and once during spring (mid-January until June) (instead of three workshops annually).

### 2.3. Sampling

The staff coordinators at the ODs were contacted to receive information about the employees’ electronic mail addresses. Purposive sampling was used to choose informants from different occupations, ages, and ODs. Initially, one hundred and ten persons (ca 5–10 employees per department or group, i.e., doctors and managers, i.e., 70 nurse anaesthetists, operating room nurses, and registered nurses, 26 nurse assistants, and 14 managers and doctors) were contacted by e-mail, and a reminder was sent one week later and twice thereafter. Thirteen persons agreed to participation and seven persons (all women) declined participation, of which three had not participated in the first workshop. Due to the low response, paper invitations were left in each employees’ mail box for all employees in every department, and additional mass-invitations were sent to increase the number of participants. One additional interview was scheduled after receiving an informed consent from the informant. One interview was performed by telephone, and 13 were performed face-to-face at the hospital. The interviews, performed between one to nine months after the first workshop, were tape-recorded, lasting from 24 to 68 min (mean 40 min).

### 2.4. The Participants

Fourteen participants (three men and 11 women), i.e., seven managers (anaesthesiologists and nurse managers), and seven employees (assistant nurses, nurse anaesthetists, and trainee doctors) accepted the invitation. The participants were from five different departments and three different groups. The participants were between 32 and 63 years old (mean 50 years). The managerial group (*n* = 7) had one to 15.5 years (mean 7 years) of experience in different managerial roles, and they had been working in the organisation between one and 24 years (mean 8 years). The employees (*n* = 7) had between two and 25 years (mean 8 years) of work experience in their profession, and they had been working in the organisation between two and 25 years (mean 19 years).

### 2.5. The Interview Guide

The interview guide (Table 1) included several questions about what was known about the STAMINA model, reasons for implementation, experiences about the workshop, making the action plans, what happened after the action plans were written, and whether there was room for improvement in the continuous work with the model. Probing questions such as “please tell more” and “what do you mean by that” were used to deepen the interview and to clear any ambiguities.

### 2.6. Data Analysis

The analysis was performed using systematic text condensation by Malterud [26,27], which has its philosophy in Giorgi’s phenomenology [28,29], focusing on people’s lived world and experiences [26,27], and thematic analysis [30,31]. First, the verbatim transcribed interviews were read several times to get an overall picture. Second, preliminary themes were identified. Meaning units (passages of text that were about the topic of interest) were identified and coded. Third, condensation was performed, i.e., similar meaning units followed by their codes were grouped together and placed under each preliminary theme. Fourth was re-contextualisation, creating the final themes, and writing the content for them. The interviews were read once more, this time with the themes in mind. Author EA performed all the steps in the analysis. In order to confirm the results, authors TH and MS read six of the interviews and the results. The final themes were a result of several discussions between the three authors.

#### Ethical Considerations

The Swedish Ethical Review Authority approved the study (Dnr 2019-00948), and informed consent was collected prior to the interviews. The study follows the regulations in the Declaration of Helsinki [32] and the local ethical guidelines and regulations [33].

## 3. Results

Five themes were identified, as presented below with citations. Scepticism and passivity turned to a more active and positive attitude by recognising the increased role of the employees. Both employees and their managers highlighted the importance of employees’ increased responsibilities in working with work environment issues.

### 3.1. Limited Knowledge of the Model and the Implementation Process

The participants described having limited information about the model and the implementation process. A common description was not knowing much about the STAMINA model before using it. Several did not remember having any information about the model, or if they remembered, they could not recall what was said. Sometimes, it was difficult to describe the goal for, or the reason why the STAMINA model was used. However, they described that there was some information, stating that *‘it was a new way to estimate one’s working environment’* (Interview 7, employee) after the reorganisation.
*‘I didn’t know much at all until we were educated on it. I don’t remember but on occasion someone told me that it would be about the work environment… ’*.(Interview 1, employee)

The participants could not describe the goal for the model or how the model was going to be implemented. They described that the model was implemented to improve the work environment due to lack of staff, difficulties in recruiting, or due to staff not feeling good. However, it made the understanding easier as the model was based on research and it gave a further seriousness to the whole process.

### 3.2. Scepticism, Lack of Confidence in the Model, and Passive Attitude

The theme describes an organisational change fatigue towards the new change. Due to earlier experiences with projects failing, there was scepticism towards the STAMINA model, as earlier and similar models had not shown any improvement. However, despite the scepticism, the participants described feelings of having given up and “accepting” to follow the process. The organisational change fatigue was described as *‘a fatigue in the organisation, where you don’t come up with suggestions for improvements anymore … I have tried to say so many times, but nothing happens…’* (Interview 9, employee).

Employees described that they got tired and gave up when they felt that their voices were not being heard. Questions about whether it would take a long time for the model to have effect could lead to negative feelings. It was too long between the workshops, but at the same time, having workshops more often seemed to be a challenge. Expectations were high about this model, and at the same time, there was a concern about whether it would *‘amount to nothing’* (Interview 10, manager) or *‘dwindle out’* (Interview 13, manager).

The nurse managers experienced scepticism and concerns before the first workshop. They described having doubts about participating in the workshop. They expected the workshop to be a place to complain about the nurse manager, and it turned out to be the opposite.
*‘…I was probably a little stressed before that (the first workshop), that I would somehow defend myself in my managerial role over what the employees experience is not good …. but it didn’t turn out that way …’*.(Interview 6, employee)

### 3.3. The Model Offered Increased Participation

The theme described the participants’ positive reactions towards increased participation in workshops as everyone could express their own thoughts. Many were surprised that even those who were not always open with their opinions could talk freely during the workshop based on the *‘uncensored and verbatim text’* (Interview 5, manager) that the employees wrote in the survey prior to the workshop. Just talking about one’s frustrations was experienced to lead to a better work environment.

The presence of the manager during the workshop made participants feel as though they *‘were being listened to’* and *‘it was better than I really expected…’* (Interview 6, employee). This increased feelings of participation and involvement in matters concerning one’s work environment.

It was also positive that problems were identified by the group of employees themselves and not *‘top down controlled’* (Interview 10, employee) *‘because if problems existing among the employees change and improve, then production increases by itself…’* (Interview 2, employee).

As the employees faced the problems every day, they could come up with solutions as well. Not everything could be solved by the employees. However, working under *‘structured forms’* (Interview 3, manager) was viewed as positive. Here, problems were identified and divided between those that could be managed within the group of employees and those that had to be sent to managers higher up in the organisation.
‘I thought the workshop was extremely positive and that as regular employees, we had a good structured session to address our concerns. There are a number of concerns, and we documented them and sat together and had an outside supervisor…’.(Interview 6, employee)

### 3.4. The Culture in the Organisation—to Understand One’s Role as Employee and Manager

The theme describes the views about the roles of employees and managers, and a willingness to change the existing culture. The participants felt that not everyone among the employees knew what was expected of them, and they experienced a culture of criticism instead of taking part and doing something about matters they were critical of. The hope was that the culture and tradition would change, and this required *‘a paradigm shift’* (Interview 12, manager). The participants expected an *‘awakening’* (Interview 4, manager) or *‘an increased consciousness’* (Interview 14, employee) from the model, especially for the employees regarding their role and increased responsibilities.
*‘to realise their (the employees) own value and responsibility for the department … to want to get involved … that I (the employee) contribute to this … I think you get this awakening that it’s not the manager who fixes everything. Maybe, I am (the employee) the one who must fix…’*.(Interview 4, manager)

The managers were described as being the important actors if the model was going to work, as *‘one follows a leader…’* (Interview 1, employee), but that at the same time, the employees could have a more active role themselves. Discussions would be *‘more open’* (Interview 6, employee) if it was led by the employees themselves. To achieve that goal, it was important for the managers in the organisation to *‘place the responsibility on the employees…’* (Interview 12, manager).

The support from the head of the department and the employees themselves was also described as essential for the model to survive as one participant described, *‘then, there is a greater chance that the model will work’* (Interview 1, employee).

### 3.5. Endurance and Feedback are Key to Success in the Implementation Process

The theme describes the need for continuity in the process of implementation. Consistency and continuation with *one* model as well as feedback given in a timely manner about how far the process has come during the initial phase of implementation were essential for the sustainability of the model. It was important to make the documented plans visible to the employees, and using different methods such as *‘pop-up information in every employee’s e-mail’* (Interview 1, employee) or just *‘information on a paper’* (Interview 7, employee) were suggested. Constantly changing a model could make the evaluation difficult. Having proper time, *‘not just 10 min’* to discuss at a deeper level in small groups that included employees from different categories, and setting aside limited time to discuss a topic were important. Moreover, participants requested a tool for problem solving or a communication tool.

Giving feedback about any improvement even in *one* question, with the goal of *‘quality over quantity’* (Interview 5, manager), and acknowledging progression in the process could give hope. It was important that someone was working with a plan *‘during the whole year’* (Interview 5, manager). Doing workshops just twice a year was too little, as one had to start over again every time. Feedback should therefore be *‘recurring point in the workplace meetings’* (Interview 5, manager) at the department.
‘That one of the problems is resolved; it’s enough, it’s phenomenal. Just think, an 8% improvement, I will be satisfied…’.(Interview 6, employee)

Even if there were problems that the employees could not address or solve themselves, these were sent to managers higher up in the organisation. To have a sustainable plan in how these documents were being handled in the organisation was described as important to the managers, and feedback about these was important to both the employees and their managers.

## 4. Discussion

The findings indicated scepticism, organisational change fatigue, and feeling of limited knowledge towards implementation of the STAMINA model, before working with it. The concept of change fatigue has been studied in nursing literature. In a literature review, McMillan and Perron [34] discussed the concepts of change fatigue, a feeling of being powerless about the change processes affecting the employees’ work life, and that the employees passively accept the implemented changes [35]. McMillan and Perron [34] claimed that there is no literature describing the benefits for change fatigue. On the other hand, change resistance is positive for the organization “in small doses” as it stimulates critical thinking. Furthermore, those who make resistance are known, in contrast to change fatigue. In the present study, it was found that, after the workshops, which increased employee participation, the participants started to realise the importance of using the model, and the employees’ increased responsibilities. They suggested that sticking to the model and giving feedback about the progression of the process were important. These findings might be interpreted as their readiness for change increased as they saw potential benefits if working according to the model.

Perhaps one of the most interesting findings in this study was the change of attitude by both the employees and their managers towards the STAMINA model and their own responsibilities, from a passive to a more active attitude. Even though there was scepticism, criticism, and disbelief towards the “new change”, none of the employees showed neither resistance nor own initiatives. This could be explained by the employees not believing in their own power in the organisation. Tetef [36] pointed out that organisations should have a plan and invest in increased awareness of each other’s knowledge to improve involvement and decision-making. Furthermore, limited knowledge or lack of information about the model generated misunderstandings about why the model was to be used and one’s own roles, which was also confirmed by Rasmussen, Hojberg, Bengtsen, and Jorgensen [21]. This might be one aspect that influenced the passive attitude that was experienced in the beginning of the process. Difficulties with creating commitment and providing information in connection to the start of using the STAMINA model has also been shown in previous research from Swedish municipalities. Even though the facilitators that are supporting the implementation receive education and perceive the model as easy to understand and provide, they also identified difficulties in transferring this information to managers and employees [37]. Furthermore, Rasmussen, Hojberg, Bengtsen, and Jorgensen [21], and Tetef [36] indicated that education should focus on the important parts that are included in an implementation process, which may have been poorly understood by the participants in the current study, as participants had difficulties to recall. One important part of the process is understanding the goal and one’s role, which is regulated in the Swedish Work Environment Authority [15,16]. Education is one way the organisations can better train the employees for the upcoming change and the new model. The participants were uncertain about whether or not they had received any information about the progression in the process. They suggested that by giving feedback and showing even small steps in progress of implementation, as was discussed by Hellman, Molin [22], the implementation process could move forward smoothly. Communication is an important tool when used properly. Sometimes organisations “deliver” information without knowing how it has been received by the recipients. This is another way of improving the understanding for the upcoming change, as the participants asked for a communication tool. Tetef Tetef [36] and Herlehy [38] stated that safe practice requires functioning communication in perioperative settings, where situational awareness about what is happening around the employees [39] prevents unpredicted events [40]. In the complex environment [41] of perioperative settings, the transfer of information and interaction between individuals in different disciplines is important. Thus, self-organisation or organising from within, is required in the implementation process [41].

Strategies to keep the model alive were discussed in one of our themes. The employees experienced that earlier when one model did not seem to work, they took on another model in order to succeed. The employees suggested that, to increase the chances of success, one should stick to *one* model and try to improve and adjust the model to the organisational needs. Jabbour et al. [42] used the term “sustainability”, pointing out education, communication, feedback, triggers, or reminders, and opportunity to have input in what is being implemented as strategies for it. In another study, the concept of “sustainability” was used [43], which points to "sense-making and value congruency", and "staff engagement and empowerment", both of which are in line with our study. The implementation process requires an ‘attractive model’, one which is relevant for one’s workplace and easy to use or modify, a goal known to all, employee engagement and support within the organisation [20,21,37]. Having a supervisor or enthusiasts according to the participants in the current study contributed to the sustainability of the model. Although it was not the focus of this study to determine the content of the action plans written after the first workshop according to the STAMINA model, several action plans have been written in different departments. A clear communication and good cooperation between different professions, how colleagues treat each other, and shortage of staff are examples of plans discussed. An earlier study indicates that implementing the STAMINA model in municipalities was a process that took time and that after a period of two years, the employees came to insight of finally shifting the focus from themselves to the organization [22].

Another interesting finding was the influence of culture and tradition that existed in the organisation in the current study. It made the participants wonder when the change in work environment was to come and who was going to take responsibility for it. Earlier, it has been discussed that several factors should work together, and prerequisites should be given to the organisation during the implementation in order to be successful, such as role definition [21,22], engaged employees [21], a strong leadership. Moreover, perioperative settings have the challenge of allowing for time to perform activities other than patient care. Production is prioritised, as delays during the work day may cause overtime work for the employees. Delays may also cause suffering for a third party, the patient, if planned surgeries are cancelled [44]. This production pressure, and not being able to stop the ongoing surgeries, to gather employees from different parts of the organisation to sit in peace and quiet may be perceived as not listening to the employees in the organisation. A change of traditions, a clear goal, a clear role definition for both the employees and their managers, and communication thereof in the organisation are essential for new changes to be successful, and for the implementation process [21,22].

### Strengths and Limitations

Several attempts were made to recruit participants from different parts of the organisation without any success. First five to 10 employees per department or group, i.e., doctors and managers, both men and women, were chosen and contacted several times. Not being successful on recruiting participants, the decision was made to send an invitation to all employees. Perhaps, this lack of interest mirrors the employees experiencing organizational change fatigue [34] or not believing in a new model. Inviting a large number of persons to participate may result in persons with special interested or extremely positive or negative in using the model to participate. However, the results of the study describe a variation, both positive and negative, and were described by citations from the participants experiences of the STAMINA model which increases the confirmability [45] in the study. Therefore, the study is interesting and gives valuable insights to the phenomena under study. According to Malterud et al. [46], it is more important to discuss “information power” which depends on study aim, sample specificity, and quality of the dialogue among others. The study aim refers to narratives and experiences of the participants. Specificity of the sample refers to a less extensive number of individuals that have specific characteristics who can provide information about the aim of the study, and finally, quality of the dialogue or the richness in the dialogue that provides the researcher with the specific information that he or she is looking for. Instead of discussing power as in quantitative studies, in qualitative studies the concept of “saturation” is discussed, which describes when no new information is found or when data repeats itself. Here, saturation was met after six interviews. However, the rest of the interviews were analysed to make sure that no new information could be found in the data.

Author EA had a pre-understanding of the perioperative context, being a nurse anaesthetist and one of the supervisors implementing the model. Being familiar with the phenomenon under study increases the credibility and confirmability of the study [45]. To avoid author EA’s pre-understanding from interfering with the findings, authors TH and MS with limited knowledge about the perioperative settings analysed six of the interviews. Special attention was paid to the topic of pre-understanding and discussed by all authors to ensure the objectivity in the results. Being one of the supervisors and conducting the interviews, there was a risk of influencing the participants. Participants talked freely both about negative and positive aspects of the implementation, despite knowing the person who interviewed them. To increase credibility and transferability, the procedure and data analysis were described as clearly as possible.

## 5. Conclusions

Scepticism turned to positive attitude by seeing that the STAMINA model offered increased participation. In order to be successful in implementation, the organisational culture must be taken into consideration by giving the employees increased responsibilities, and timely feedbacks. Role description, goal definition, feedback, and sticking to one model are key factors for success.

## Figures and Tables

**Table 1 ijerph-17-03037-t001:** The interview guide.

**Demographic Data**	Gender (male, female) Age Current workplace Number of employees (the question is for nurse managers and leaders) Number of years in the current workplace/in the profession/as nurse manager
**Main Questions**	Knowledge about the model • What did you know about the STAMINA model before it was implemented? • What information did you receive, and did you know why this should be implemented? • How did you experience the workshop? What were your first thoughts about the workshop? • What were your expectations and fears before the workshop itself? • Do you have any experience with similar initiatives such as the STAMINA model? • What are your thoughts on previous initiatives in relation to the STAMINA model? • How do you feel that the STAMINA model differs from systematic work environment management that you have done before? • What did you expect from the STAMINA model? • In what way do you think the workshop can/should contribute to a better work environment? • What were your concerns about the workshop? Implementation of workshop • How did the work progress with the workshop? • Did the report (according to the model) contribute to the workshop implementation? • What was positive and negative about the workshop? Please give concrete examples. • What was the mood of the group during the workshop? Was everyone given the opportunity to speak? • Were your expectations met by the workshop? Why /why not? Thoughts following the intervention/workshop • What is your experience of the result of the workshop about employee responsibilities and the relationship with the managers? • Action plans - have they made any changes yet? What? Please tell more. • How many and what kind of meetings have you had after the workshop to work on the action plans? • How do you work with the action plans? (Who is responsible for follow-up? What role do the manager and employees have?) Please give a concrete example. • What is missing in this model according to you? • What do you/your employees expect from following up on the workshop? • Please tell us whether working with the workshop has contributed to work with environmental questions in your unit or not?
**Probing Questions**	Please tell more. What do you mean by that?

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
