# Peer review of "Experiences of the Initial Phase Implementation of the STAMINA-Model in Perioperative Context Addressing Environmental Issues Systematically—A Qualitative Study"

_ijerph, 2020, doi:10.3390/ijerph17093037_

Round 1

Reviewer 1 Report

General comments: This manuscript presents some interesting findings, but the concepts and terminologies discussed are difficult to grasp and require further elaboration. Authors should work on the readability of the manuscript. Specific comments: 1. Please change all abbreviations "Stamina" to UPPERCASE "STAMINA" as per the title. 2. Was the study protocol prospectively registered? 3. How was sample size determined? There is currently no evidence of power calculation. The present sample appears small and limited to a convenience sample. Is this was meant to be a pilot trial, please specify. 4. What is "organisational tiredness"? Do you mean "organizational change fatigue", which is a general sense of apathy or passive resignation towards organizational changes by individuals or teams? Does this reflect the low recruitment rates? 5. What is the baseline attrition rate of this hospital compared to other hospitals in Sweden? Is it comparable, higher or lower? 6. What qualifies as "survival of the model"? This requires further substantiation.

Author Response

Editor-in Chief             200403                                                                           

Professor Fiona Timmins

Journal of Environmental Research and Public Health

Dear Reviewer,

We thank the you for your valuable comments that we hope have improved the manuscript. We have revised the manuscript according to the concrete suggestions we received. Our responses are itemized below. We now hope that the revision is to satisfaction and that the manuscript now can be accepted for publication but are willing to do further changes if required.

Yours Sincerely

The authors

General comments: This manuscript presents some interesting findings, but the concepts and terminologies discussed are difficult to grasp and require further elaboration. Authors should work on the readability of the manuscript.

Thank you for your comment. We have revised the manuscript according to your comments. However, we are happy to revise this again if there are further unclarities.

Specific comments:

  1. Please change all abbreviations "Stamina" to UPPERCASE "STAMINA" as per the title. Thank you for your comment, we have changed it throughout the entire text.

  1. Was the study protocol prospectively registered? The study protocol refers to a larger project which preceded this particular study. This has been clarified in the manuscript in lines 115-119.

“This study is a continuation of a larger project conducted in municipalities in Sweden. A description of the larger project is to be found in a prospective designed protocol [1]. This protocol also includes a detailed description of the STAMINA model. In the current study, after ethical approval by the Swedish Ethical Review Authority, a qualitative prospective study design was conducted in the new context of perioperative settings to capture the experiences of the self-sustaining implementation process of the support model and how it was perceived that the model contributed to the goal of a better working environment. “

  1. How was sample size determined? There is currently no evidence of power calculation. The present sample appears small and limited to a convenience sample. Is this was meant to be a pilot trial, please specify.

Thank you for your comment. In qualitative studies we do not use power. However, other measures are used to decide whether the number of participants are enough or not. According to Malterud and colleagues [2] it is more important to discuss information power which depends on study aim, sample specificity and quality of the dialogue among others. Study aim refers to narratives and experiences of the participants. Specificity of sample refers to less extensive number of individuals that have specific characteristics who can provide information about the aim of the study, and finally quality of the dialogue or the richness in the dialogue that provides the researcher with the specific information that he or she is looking for. Instead of discussing power, in qualitative studies the concept of “saturation” is discussed, which describes when no new information is found or when data repeats itself. Then the researcher knows that saturation has been met and that the number of participants has been sufficient. Studies have been conducted with similar sample sizes earlier [3].

We have clarified this in the methodological discussion section in lines 383-392.

  1. What is "organisational tiredness"? Do you mean "organizational change fatigue", which is a general sense of apathy or passive resignation towards organizational changes by individuals or teams? Does this reflect the low recruitment rates? Yes, thank you for your suggestion. We have changed “organisational tiredness” to “organizational change fatigue”. This may explain the low recruitment rates as you mention. We have expanded our discussion and methodological discussion adding the information above. Please see lines 212, 216, 295, 293, 297-303, 377.

  1. What is the baseline attrition rate of this hospital compared to other hospitals in Sweden? Is it comparable, higher or lower? Quits and separations? Sick days? General overwork? Expressions of hostility?

Thank you for your comment. The study was not conducted in the entire hospital, but in the perioperative settings. These are interesting questions. The Human Resources Department of at the hospital under study indicated that the attrition rate of the hospital was lower compared to other university hospitals in Sweden, but that the percentage of quits was higher in comparison to others. We have added this information in lines 124-126.

Extra information given below we prefer not to ad that into the manuscript

For the department of Anaesthesia, Operations and Intensive care where department of Anaesthesia and Operations is includes the sick leave was between 6-7% during the last year (long time sick leave which is more than 60 days were 3.27%) and approximately 54% of women and 58% of men (new employees n=1641) were in good health and present at work during 2019. Employee turnover was 4.18% for the year 2019.    

  1. What qualifies as "survival of the model"? This requires further substantiation.

Thank you for your comment. We understand that the wordings “survival of the model” might be difficult to grasp. Actually, what we aim at is sustainability and we have now changed the wordings in the manuscript. We hope that this clarification will make the text easier to understand. Please see lines 274 and 345-347, and 351.

References

  1. Svartengren, M. and T. Hellman, Study protocol of an effect and process evaluation of the Stamina model; a Structured and Time-effective Approach through Methods for an Inclusive and Active working life. BMC Public Health, 2018. 18(1): p. 1070.
  2. Malterud, K., V.D. Siersma, and A.D. Guassora, Sample Size in Qualitative Interview Studies: Guided by Information Power. Qual Health Res, 2015.
  3. Hellman, T., F. Molin, and M. Svartengren, A Mixed-method Study of Providing and Implementing a Support Model Focusing on Systematic Work Environment Management. J Occup Environ Med, 2020.

Reviewer 2 Report

This paper seeks to engage in a qualitative evaluation of the STAMINA model, a model for engaging employees in improving work culture among perioperational employees at Swedish hospitals. The hospital in question implemented the STAMINA model widely, and the researchers sought to understand how the model was accepted by front-line employees and healthcare managers. The results suggest slow but steady increase in buy-in to the program and some belief that tangible results would be possible.

The project has a number of strengths; (1) the interview protocol is nicely laid out, so we know what the queries and prompts are to respondents along with follow-up questions; and (2) there is cross-validation of the coding of responses by the authors on the project.

The paper suffers from three weaknesses, one that could be fixed easily and another two that will require more work;

  • On the easy side, we need more information regarding what the exact working conditions are like in the perioperational settings and what the specific issues are that need to be addressed in the specific setting where the research is being done. The national government mandate suggests that Swedish authorities believe that work environments are important, but that doesn’t really describe what the specific working conditions are like – is there a high degree of burnout? Quits and separations? Sick days? General overwork? Expressions of hostility? Etc. This would provide an important context for interpreting the results.
  • The number of interviews conducted (14) relative to the initial sample size is a serious concern. There is a serious possibility of sample selection here where interviewees are either unusually optimistic that the STAMINA program will work or unusually pessimistic. Alternatively, those who are most disengaged might not volunteer for the study at all. Some basic comparison of how the respondents compare to the other potential interviewees in terms of job title, race/ethnicity, gender, job tenure, etc. would be helpful here. The low number isn’t as big of a problem if the interviewees are somewhat representative of the unit.
  • Finally, in the conclusion, it would be nice to know what organizational changes were tried as a result of the input from employees and managers into the STAMINA program. Even if there is no way to tell if these interventions worked, it would nice to tie what the organization was trying to do with the perceived problems discussed in (1).

Good luck with the research.

Author Response

Editor-in Chief                             200408

Professor Fiona Timmins

Journal of Environmental Research and Public Health

Dear Reviewer,

We thank the you for your valuable comments that we hope have improved the manuscript. We have revised the manuscript according to the concrete suggestions we received. Our responses are itemized below. We now hope that the revision is to satisfaction and that the manuscript now can be accepted for publication but are willing to do further changes if required.

Yours Sincerely

The authors

Specific Comments

  • Comment 1- On the easy side, we need more information regarding what the exact working conditions are like in the perioperational settings and what the specific issues are that need to be addressed in the specific setting where the research is being done. The national government mandate suggests that Swedish authorities believe that work environments are important, but that doesn’t really describe what the specific working conditions are like – is there a high degree of burnout? Quits and separations? Sick days? General overwork? Expressions of hostility? Etc. This would provide an important context for interpreting the results.

Thank you for your comment. We have added more information about working conditions, reasons for wanting to leave one’s work and psychosocial work environment in the background under the title “The perioperative environment and its demands” and under “Study Settings”.

The perioperative environment and its demands (lines 50-62)

Vowels, Topp [1] indicated that operating room nurses experienced “pressure to work quickly” as a stressful event in their everyday work where patient turnover is high. According to Logde, Rudolfsson [2] specialist nurses in perioperative settings (i.e. nurse anaesthetists and operating room nurses) leave their work due to rough psychosocial environment and systematic freezing out colleagues, poor wage growth, dissatisfactory work schedule, working night and day shifts, and a nurse manager who is described as absent, non- including and “nonchalant”. In other study emotional exhaustion Lee, MacPhee [3] and Sillero and Zabalegui [4] depersonalization, and reduced personal accomplishment was found among nurses in perioperative context. The study by Lee, MacPhee [3] pointed to nurse manager ability, leadership, and support of nurses, staffing and resources adequacy; and nursing foundations of quality care as factors affecting the occurrence of burnout. Moreover, in perioperative context physician to nurse violence [5] and nurse to nurse violence [6] mong operating room nurses have been described earlier.”

Study Settings

Lines 124-126

The attrition rate of the nurses in the hospital under study was lower compared to other university hospitals in Sweden, but that the percentage of quits was higher in comparison to others.

Lines 131-137

The working conditions in the perioperative settings are unique. The operating rooms are closed rooms, sometimes without access to daylight. Patient safety and hygiene reasons lead to specialist nurses, OR-nurses, not leaving the operating room while working with the patient. Patient care demands high concentration. During day shifts, only small lunch break of 30–45 min and two small coffee breaks of 10–15 min each are taken by the specialist nurses. OR- Nurses take usually their breaks in between two patients. High demands are placed on both specialist nurses because of these working conditions during long hours of work [2].

We are happy to revise this section again if there are further unclarities.

Here are some answers to your questions:

Extra information given below we prefer not to ad that into the manuscript

For the department of Anaesthesia, Operations and Intensive care where department of Anaesthesia and Operations is includes the sick leave was between 6-7% during the last year (long time sick leave which is more than 60 days were 3.27%) and approximately 54% of women and 58% of men (new employees n=1641) were in good health and present at work during 2019. Employee turnover was 4.18% for the year 2019.    

  • Comment 2- The number of interviews conducted (14) relative to the initial sample size is a serious concern. There is a serious possibility of sample selection here where interviewees are either unusually optimistic that the STAMINA program will work or unusually pessimistic. Alternatively, those who are most disengaged might not volunteer for the study at all. Some basic comparison of how the respondents compare to the other potential interviewees in terms of job title, race/ethnicity, gender, job tenure, etc. would be helpful here. The low number isn’t as big of a problem if the interviewees are somewhat representative of the unit.

Thank you for your comment. We are aware of, and understand, your concerns regarding the recruitment procedure and fully agree with you. However, as the findings include a variety of experiences of the initial implementation of the STAMINA model we do believe that we have included participants that not only are positive or negative in their opinions. In our final sample we also include participants from various departments and that have been clarified in the manuscript. To show that the participants were from five different ODs and three different groups we have added information in line 163-164. In line 153 we have added the information about the persons who declined participation (all were women).

In lines 150-151 we have added information about the 110 persons who were asked to participate: “70 nurse anaesthetists, operating room nurses and registered nurses, 26 nurse assistants, and 14 managers and doctors”.

We have expanded the methodological discussion and about the sample size there is additional argumentation in lines 383-392.

  • Finally, in the conclusion, it would be nice to know what organizational changes were tried as a result of the input from employees and managers into the STAMINA program. Even if there is no way to tell if these interventions worked, it would nice to tie what the organization was trying to do with the perceived problems discussed in (1).

Thank you for your comment. Although it was not the focus of this study to determine the content of the action plans written after the first workshop according to the STAMINA model, several action plans have been written in different departments. A clear communication and good cooperation between different professions, how colleagues treat each other, and shortage of staff are examples of plans discussed. An earlier study indicates that implementing the STAMINA model in municipalities was a process that took time and that after a period of two years the employees came to insight of finally shifting the focus from themselves to the organization [7]. Please find our clarification in discussion section, lines 351-357.

References

  1. Vowels, A., R. Topp, and J. Berger, Understanding stress in the operating room: a step toward improving the work environment. Ky Nurse, 2012. 60(2): p. 5-7.
  2. Logde, A., et al., I am quitting my job. Specialist nurses in perioperative context and their experiences of the process and reasons to quit their job. Int J Qual Health Care, 2018. 30(4): p. 313-320.
  3. Lee, S.E., M. MacPhee, and V.S. Dahinten, Factors related to perioperative nurses' job satisfaction and intention to leave. Jpn J Nurs Sci, 2020. 17(1): p. e12263.
  4. Sillero, A. and A. Zabalegui, Organizational Factors and Burnout of Perioperative Nurses. Clin Pract Epidemiol Ment Health, 2018. 14: p. 132-142.
  5. Higgins, B.L. and J. MacIntosh, Operating room nurses' perceptions of the effects of physician-perpetrated abuse. Int Nurs Rev, 2010. 57(3): p. 321-7.
  6. Bigony, L., et al., Lateral violence in the perioperative setting. Aorn j, 2009. 89(4): p. 688-96; quiz 697-700.
  7. Hellman, T., F. Molin, and M. Svartengren, A Qualitative Study on Employees' Experiences of a Support model for Systematic Work Environment Management. Int J Environ Res Public Health, 2019. 16(19).

Round 2

Reviewer 1 Report

  1. Please change "the percentage of quits was higher" to "the percentage of personnel who resigned was higher".
  2. Please check your references. Citation style appears inconsistent and ref [17] is incomplete.

Author Response

Department of Surgical Sciences

Uppsala University

Journal of Environmental Research and Public Health

Dear Reviewer,

We thank the you once again for your valuable comments. We have revised the manuscript according to your suggestions we received. Our responses are itemized below. We now hope that the revision is to satisfaction but are willing to do further changes if required.

Yours Sincerely

The authors

  • Please change "the percentage of quits was higher" to "the percentage of personnel who resigned was higher".

The sentence ”the percentage of quits was higher" was replaced with “but that the percentage of personnel who resigned was higher in comparison to others”. Please see lines 126-127.

  • Please check your references. Citation style appears inconsistent and ref [17] is incomplete.

Reference 17 was re-examined in lines 467-469 and corrected. We looked through the citation style and all the references again, this time with the help of a librarian.
